# Analog Schwarzschild Black Hole from a Nonisentropic Fluid

**Neven Bilić * and Hrvoje Nikolić**

Theoretical Physics Division, Rudjer Bošković Institute, 10002 Zagreb, Croatia; hnikolic@irb.hr
* Correspondence: bilic@irb.hr

**Abstract:** We study the conditions under which an analog acoustic geometry of a relativistic fluid in flat spacetime can take the same form as the Schwarzschild black hole geometry. We find that the speed of sound must necessarily be equal to the speed of light. Since the speed of the fluid cannot exceed the speed of light, this implies that analog Schwarzschild geometry necessarily breaks down behind the horizon.

**Keywords:** analog gravity; Schwarzschild metric; nonisentropic fluid

## 1. Introduction

The analog acoustic metric was first introduced by Unruh [1], with the motivation of explaining the Hawking radiation produced by black holes [2]. Since then, extensive research on analog gravity has been done (see, e.g., [3,4] for reviews), but analog black holes do not seem to have shed much light on one of the most difficult problems with Hawking radiation—the black hole information paradox [5–13].

In general, the analog acoustic metric $G_{\mu\nu}$ takes the form [14]

$$G_{\mu\nu} = \omega[g_{\mu\nu} - (1 - c_\text{s}^2)u_\mu u_\nu], \tag{1}$$

where $u_\mu$ is the 4-velocity of a relativistic fluid in the background metric $g_{\mu\nu}$ usually taken to be the flat Minkowski metric, $c_\text{s}$ is the speed of sound, and the conformal factor $\omega$ is related to the equation of state of the fluid. Normally, the fluid is assumed to satisfy the Euler equation without external pressure, and particle number conservation is assumed.

Unfortunately, with these assumptions, the acoustic metric of the form (1), which mimics the Schwarzschild black hole, cannot be found. It has been noted that a non-relativistic version of the metric (1) can be found, which differs from the Schwarzschild metric by a non-constant conformal factor [3,4,15]. However, even in this case, one must assume an external force field to satisfy the Euler equation. De Oliveira et al. [16] have recently obtained a closed form of an analog Schwarzschild geometry in a setup with an external force. Unfortunately, their solution is subject to a slightly modified analog geometry that is not exactly supported by fluid dynamics. For nonisentropic fluids, recently studied in [17], one is more flexible and has the possibility to choose $\omega(x)$, such that the analog metric exactly reproduces the Schwarzschild metric. Since $g_{\mu\nu}$ is the flat metric that is not proportional to the Schwarzschild metric, one would naively expect $G_{\mu\nu}$ to be equal to the Schwarzschild metric only if $c_\text{s}$ in (1) is not equal to unity.

On the contrary, we find here that $G_{\mu\nu}$ can be equal to the Schwarzschild metric if $c_\text{s} \to 1$, so that $(1 - c_\text{s}^2)u_\mu u_\nu$ remains non-zero. Since the acoustic horizon is by definition a surface beyond which the fluid is faster than the speed of sound, it follows that the fluid beyond the horizon should be superluminal, which is physically forbidden. This means that analogue metric can take the form of a Schwarzschild geometry only outside of the black hole, and not in the black hole interior. The horizon thus represents a physical boundary beyond which the Schwarzschild geometry necessarily breaks down. This is consistent with the expected behavior of black hole firewalls [18,19].

The remainder of the paper is organized as follows. In Section 2, we derive the Schwarzschild geometry as an analog gravity model in a relativistic fluid. In Section 3, we briefly discuss the field theoretic description of the fluid relevant to the model considered in Section 2. The concluding section, Section 4, is devoted to discussion and conclusions.

## 2. Analog Schwarzschild Geometry

In applications of analog geometry, particle number conservation is usually assumed, in addition to energy-momentum conservation or the Euler equation. However, with this assumption, some interesting geometries cannot be mimicked by analog geometry. The fluid in which the particle number is not conserved is generally nonisentropic.

Here we study the conditions under which one can mimic the Schwarzschild geometry. We assume the metric that is conformally equivalent to the Schwarzschild metric, i.e.,

$$ds^2 = \omega(r,t)\left[\gamma(r)dt^2 - \gamma(r)^{-1}dr^2 - r^2 d\Omega^2\right], \tag{2}$$

where

$$\gamma(r) = 1 - \frac{2MG}{r}. \tag{3}$$

We look for an analog fluid model that mimics the metric of the form (2), and we impose the necessary conditions so that the conformal factor $\omega$ can be set to one. The basic idea is to find a suitable coordinate transformation $t \to \tilde{t}$, such that the new metric takes the form of the relativistic acoustic metric

$$G_{\mu\nu} = \frac{n}{m^2 c_s \omega}[g_{\mu\nu} - (1 - c_s^2)u_\mu u_\nu], \tag{4}$$

where $g_{\mu\nu}$ is the Minkowski metric in spherical coordinates, $u_\mu$ is the four-velocity with non-vanishing radial component, $m$ is an arbitrary mass scale, $n$ is the particle number density, and $w$ is the specific enthalpy defined as

$$w = \frac{p + \rho}{n}. \tag{5}$$

We assume that the fluid is irrotational and satisfies Euler's equation. Accordingly, we assume that the enthalpy flow $wu_\mu$ is a gradient of a scalar potential, i.e.,

$$wu_\mu = \theta_{,\mu}, \tag{6}$$

and that the entropy gradient is proportional to the gradient of $\theta$ [17]

$$s_{,\mu} = \frac{u^\nu s_{,\nu}}{w}\theta_{,\mu}. \tag{7}$$

Next, following [20], we apply a coordinate transformation

$$t = \kappa\tilde{t} + f(r), \tag{8}$$

where $\kappa$ is a constant, and the function $f(r)$ is to be determined by the condition that the transformed metric is of the form (4). The line element (2) in the new coordinates is of the form

$$ds^2 = \omega(r,t)\left[\kappa^2\gamma(r)d\tilde{t}^2 + 2\kappa\gamma f'd\tilde{t}dr - \left(\gamma(r)^{-1} - \gamma(r)f'^2\right)dr^2 - r^2 d\Omega^2\right], \tag{9}$$

similar to the Painlevé–Gullstrand metric. Comparing this with (4), we obtain a set of equations

$$\kappa^2\gamma = 1 - (1 - c_s^2)u_{\tilde{t}}^2, \tag{10}$$

$$\kappa \gamma f' = -(1 - c_s^2) u_{\tilde{t}} u_r, \tag{11}$$

$$\left( \gamma(r)^{-1} - \gamma(r) f'^2 \right) = 1 + (1 - c_s^2) u_r^2, \tag{12}$$

$$u_{\tilde{t}}^2 - u_r^2 = 1, \tag{13}$$

and we require that the conformal factor in (2) is equal to that of (4); that is,

$$\omega = \frac{n}{m^2 c_s w}. \tag{14}$$

Equations (10)–(13) give

$$c_s = \kappa, \tag{15}$$

$$u_{\tilde{t}} = \frac{(1 - \kappa^2 \gamma)^{1/2}}{(1 - \kappa^2)^{1/2}}, \quad u_r = -\frac{\kappa(1 - \gamma)^{1/2}}{(1 - \kappa^2)^{1/2}}, \tag{16}$$

$$f' = \frac{(1 - \kappa^2 \gamma)^{1/2}(1 - \gamma)^{1/2}}{\gamma}. \tag{17}$$

We could absorb $\kappa$ in the $\tilde{t}$-coordinate in (9) and formally set $\kappa$ to a suitable constant in (17), e.g., $\kappa = 0$ or $\kappa = 1$. With $\kappa = 0$, we would obtain the line element in the form

$$ds^2 = \frac{n}{m^2 c_s w} [\gamma d\tilde{t}^2 + 2(1 - \gamma)^{1/2} d\tilde{t} dr dr^2 - r^2 d\Omega^2], \tag{18}$$

conformally equivalent to the Painlevé-Gullstrand metric. With $\kappa = 1$, we would obtain

$$ds^2 = \frac{n}{m^2 c_s w} [\gamma d\tilde{t}^2 + 2(1 - \gamma) d\tilde{t} dr - (2 - \gamma) dr^2 - r^2 d\Omega^2]. \tag{19}$$

So far, we basically agree with previous works [3,4,15], which have rendered the metric conformally equivalent to the Schwarzschild metric in Painlevé-Gullstrand coordinates. We differ only in the conformal factor, since these papers use a non-relativistic acoustic metric.

From now on, we depart from the approach of Refs. [3,4,15], in which the continuity equation is imposed, and the external force is invoked to preserve the consistency of the definition of the speed of sound with the Euler equation. Instead, we adopt the approach of Ref. [17], which is different from the approaches in Refs. [3,4,15] basically, in two assumptions. First, we do not require isentropy of our fluid, and therefore the continuity equation need not be imposed. Second, we adhere to the standard definition of the the speed of sound without invoking any external force.

By applying the potential-flow Equation (6), we derive closed-form expressions for $w$, $n$, and $\omega$. Since the metric (9) is stationary, except for the conformal factor $\omega$, the velocity potential must be of the form

$$\theta = m(\tilde{t} + g(r)), \tag{20}$$

where $g(r)$ is a function of $r$ and $m$ is an arbitrary mass. Then, Equation (6) gives

$$w = \frac{m}{u_{\tilde{t}}} = m \frac{(1 - \kappa^2)^{1/2}}{(1 - \kappa^2 \gamma)^{1/2}}. \tag{21}$$

Moreover, it follows from (6) that the function $g$ in (20) must satisfy

$$g' = \frac{w}{m} u_r = -\frac{\kappa(1 - \gamma)^{1/2}}{(1 - \kappa^2 \gamma)^{1/2}}. \tag{22}$$

From the definition of the speed of sound

$$c_s^2 = \left. \frac{\partial p}{\partial \rho} \right|_s = \frac{n}{w} \left( \left. \frac{\partial n}{\partial w} \right|_s \right)^{-1}, \tag{23}$$

where the subscript $_s$ denotes that the specific entropy $s$ is kept fixed, we find

$$n(w,s) = c_1(s)w^{1/\kappa^2},\tag{24}$$

where $c_1(s)$ is an arbitrary function of $s$. The specific entropy $s$ is generally a function of $\theta$ [17], so (20) implies $s = s(\tilde{t} + g(r))$. Then, from (14), we obtain

$$\omega = \frac{c_1(s)}{\kappa m^{3-1/\kappa^2}} \left( \frac{1-\kappa^2}{1-\kappa^2\gamma(r)} \right)^{(1/\kappa^2-1)/2}.\tag{25}$$

Clearly, if the speed of sound $c_s \equiv \kappa \neq 1$, the conformal factor is a nontrivial function of $r$ and $\tilde{t}$, so the acoustic metric in ordinary fluids with $c_s < 1$ can only describe the metric conformally invariant to the Schwarzschild metric. However, if we choose

$$c_1(s) = m^{3-1/\kappa^2}h(s)^{1/\kappa^2-1},\tag{26}$$

where $h$ is a function of $s$ independent of $\kappa$, we get $\omega \to 1$ in the ultra-relativistic limit $c_s \to 1$ of a stiff fluid, and the acoustic metric approaches the Schwarzschild metric arbitrarily close.

At first sight, it looks as if the metric (4) is ill-defined in the limit $c_s \to 1$. However, we can absorb the factor $(1 - c_s^2)$ into the velocity vector by redefining

$$u_\mu = (1 - c_s^2)^{-1/2}\tilde{u}_\mu,\tag{27}$$

with normalization

$$g^{\mu\nu}\tilde{u}_\mu\tilde{u}_\nu = 1 - c_s^2,\tag{28}$$

so that in the limit $c_s \to 1$ the vector $\tilde{u}_\mu$ becomes light-like. In that limit we have

$$\tilde{u}_{\tilde{t}} = -\tilde{u}_r = \sqrt{2MG/r},\tag{29}$$

$$f = 2MG\ln\left(\frac{r}{2MG} - 1\right).\tag{30}$$

Now, the acoustic metric

$$G_{\mu\nu} = g_{\mu\nu} - \tilde{u}_\mu\tilde{u}_\nu\tag{31}$$

is identical to the Schwarzschild metric in the form similar to Painlevé-Gullstrand coordinates with line element

$$ds^2 = \gamma d\tilde{t}^2 + 2(1-\gamma)d\tilde{t}dr - (2-\gamma)dr^2 - r^2d\Omega^2.\tag{32}$$

It may be easily checked that, by applying the coordinate transformation inverse to (8) with $\kappa = 1$ and $f$ defined in (30), the Schwarzschild metric is recovered in the standard diagonal form.

### 3. Field Theoretic Description of the Nonisentropic Fluid Flow

In general, the fluid can be described as a classical field theory in terms of a scalar field $\theta(x)$ and the kinetic term $X = g^{\mu\nu}\theta_{,\mu}\theta_{,\nu}$ through the prescription [17]: $p(w,s) = \mathcal{L}(X,\theta)$, $w = \sqrt{X}$, $n = 2\sqrt{X}\mathcal{L}_X$, where the subscript $_X$ denotes a derivative with respect to $X$. The specific entropy $s$ may be identified with $\theta$ or more generally with an unknown function of $\theta$.

From the thermodynamic relation

$$n = \left.\frac{\partial p}{\partial w}\right|_s\tag{33}$$

we have

$$p(w,s) = \int dw\, n(w,s). \tag{34}$$

Using (24), this gives

$$p(w,s) = c_1(s)\frac{w^{1+1/\kappa^2}}{1+1/\kappa^2} + c_2(s), \tag{35}$$

where the integration constant $c_2$ is a function of $s$. This gives the on-shell Lagrangian

$$\mathcal{L} = W(\theta)\frac{\sqrt{X^{1+1/\kappa^2}}}{1+1/\kappa^2} - V(\theta), \tag{36}$$

where we have denoted $c_1(s) = W(\theta)$, $c_2(s) = -V(\theta)$. The function $V(\theta)$ is not arbitrary, since we must also satisfy the equation of motion [17]

$$(2\mathcal{L}_X\, g^{\mu\nu}\theta_{,\mu})_{;\nu} - \partial\mathcal{L}/\partial\theta = 0. \tag{37}$$

Consider next the limit $1 - \kappa^2 \equiv \epsilon \to 0$. In this limit, we find

$$w = \sqrt{X} = m\epsilon^{1/2}\left(\frac{r}{2MG}\right)^{1/2}, \tag{38}$$

$$\mathcal{L} = \frac{m^{3-\epsilon}}{2}h(s(\theta))^\epsilon X X^{\epsilon/2} - V(\theta), \tag{39}$$

$$n = m^{3-\epsilon}h(s(\theta))^\epsilon\sqrt{X}X^{\epsilon/2} = m^3 h(s(\theta))^\epsilon \epsilon^{1/2}\left(\frac{r}{2MG}\right)^{1/2}. \tag{40}$$

The equation of motion (37) can be written as

$$2\partial_{\tilde{t}}(\mathcal{L}_X\partial_{\tilde{t}}\theta) - \frac{2}{r^2}\partial_r(r^2\mathcal{L}_X\partial_r\theta) = \frac{\partial\mathcal{L}}{\partial\theta}. \tag{41}$$

This, together with (20) and (22), has a leading order in $\epsilon$

$$\frac{2m^3}{r} = -\frac{\partial V}{\partial\theta}. \tag{42}$$

This equation can be solved by making use of the explicit functional relation between $\theta$ and $r$. From (20) and (22), one finds

$$\theta = -4mMG\kappa\left(\frac{r}{2MG\epsilon} + \frac{\kappa^2}{\epsilon^2}\right)^{1/2} + m\tilde{t} + \text{const.} \tag{43}$$

In the limit $\epsilon \to 0$, we may neglect the $\tilde{t}$ dependence, and keeping the leading orders in $\epsilon$, we obtain

$$r = \frac{\epsilon\theta^2}{8m^2MG} - \frac{2MG}{\epsilon}. \tag{44}$$

Then, substituting this for $r$ in (42) and integrating, we obtain

$$V(\theta) = 2m^4\ln\left|\frac{\epsilon\theta + 4mMG}{\epsilon\theta - 4mMG}\right|. \tag{45}$$

In the limit $\epsilon \to 0$, the potential tends to zero as $V \sim \epsilon\theta m^3/(2MG)$, so that the Lagrangian (39) in this limit becomes a free massless scalar field Lagrangian. This Lagrangian describes a stiff fluid with the equation of state $p = \rho$ and the sound speed equal to the speed of light, as it should according to the discussion in Section 2.

## 4. Conclusions

By making use of nonisentropic fluid dynamics, we have succeeded in modeling an analog of the Schwarzschild spacetime. Applying an analog metric in the form similar to the metric in Painlevé–Gullstrand coordinates, we have been able to reproduce the exterior of a Schwarzschild black hole in the limit when the speed of sound approaches the speed of light. In this case, the acoustic horizon is a physical boundary beyond which the Schwarzschild geometry breaks down in a way similar to the expected behavior of black hole firewalls [18,19].

It is important to stress that neither the firewall proposal nor our analog geometry model can make specific predictions for physics in the black hole interior. Nevertheless, our analog model offers a physically clear explanation as to why the interior cannot be described by Schwarzschild geometry. This explanation does not have any explicit dependence on quantum theory and small distance physics, nor does it depend on the existence of analog Hawking radiation. It only depends on the assumption that curved geometry emerges from the acoustic properties of a relativistic fluid. This result is compatible with the general idea that classical Einstein gravity could be emergent as a fluid phase of some more fundamental degrees of freedom [21,22].

It is worth mentioning that the analog metric in this paper and in other papers on analog gravity describes an effective geometry in which fluid acoustic perturbations propagate and cannot be related to the Einstein equations, as in general relativity. In particular, the classical BH thermodynamics (first, second and third law) are tightly related to the Einstein equations, and hence, these thermodynamic laws cannot be reproduced by an analog metric. However, as demonstrated in previous works, e.g., Refs. [3,4,14], the analog horizon radiation, dubbed also the analog Hawking radiation, can be reproduced in the usual way. The analog horizon radiation is a genuine quantum effect in which the phonon quanta participate.

Finally, we would like to mention that we are not aware of any astrophysical research related to the study presented here. However, in the framework of the relativistic acoustic geometry in general, possible effects could be observed in the BH accretion [23] and in high energy experiments [24,25].

**Author Contributions:** Conceptualization, N.B.; methodology, N.B. and H.N.; formal analysis, N.B. and H.N.; writing—original draft preparation, H.N.; writing—review and editing, N.B. and H.N. All authors agreed to the published version of the manuscript.

**Funding:** This research was funded by the ICTP-SEENET-MTP project NT-03 Cosmology-Classical and Quantum Challenges.

**Acknowledgments:** The work of N.B. has been partially supported by the ICTP-SEENET-MTP project NT-03 Cosmology-Classical and Quantum Challenges.

**Conflicts of Interest:** The authors declare no conflict of interest.

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
