# Peer review of "Analog Schwarzschild Black Hole from a Nonisentropic Fluid"

_universe, doi:10.3390/universe7110413_

Round 1

Reviewer 1 Report

The article deals with obtaining an acoustic metric analogous to Schwarzschild.  The theme seems to be very relevant in light of the latest research on Hawking radiation in acoustic systems analogous to black holes.  However, some points need to be clarified:

1) Is the omega factor in (2) the same as in (4)?  

2) Why is there no speed of sound in the line element (2)?  After all, the temporal dimension is not the same as the spatial one.  I believe that once this dimensionality problem is solved, the k factor will become arbitrary.

 3) It is not clear why there is no omega factor in expressions (15) and (16).  There is confusion in equation (22) with this factor.

Reviewer 2 Report

In this paper the authors find the conditions under which the Schwarzschild geometry can be described by an acoustic geometry in flat spacetime.
This gives rise to an alternative description to the black hole geometry 
in the outer region of the horizon. 

My main concern on this work is the following. As already mentioned, in this paper the authors give  an alternative description to the black hole  
in the outer region of the horizon,  this makes the result interesting.
But considering that the authors have this new interpretation there  isn't a real physical analysis of the  results(i.e. can we derive the thermodynamical properties of the
 Black Hole, are these consistent with the results obtained from the usual GR description 
 of the Schwarzschild black hole?)

Before I can recommend the manuscript for publications, I will like
the authors do a  physical analysis of the thermodynamics of the black hole in terms of the acoustic geometry.

Reviewer 3 Report

In this article (communication), analog Schwarzschild black hole from a nonisentropic fluid is argued. In particular, the conditions that an analog acoustic geometry of a relativistic fluid in flat spacetime can take the same form as the Schwarzschild black hole geometry are investigated. The discussions could be interesting and the analogy presented in this work might be helpful for the related future works. Hence, if the following points are reconsidered very carefully, this paper could be reconsidered for publication. 

1. There would exist past related works on the relation between an acoustic geometry of a relativistic fluid and that of the Schwarzschild black holes in the literature. By comparing with these preceding studies, the novel ingredients and significant progresses of this work should be stated more explicitly and in more detail. That is, the differences between this paper and the past ones should be described in more detail and more clearly. This is the most crucial point in this review. 

2. It is stated that the speed of sound must necessarily be equal to the speed of light. As a consequence, since the speed of fluid cannot exceed the speed of light, this implies that analog Schwarzschild geometry necessarily breaks down behind the horizon. From this point, what physics on black holes can be deduced in general relativity? 

3. It seems the considerations devellopped here are within the classical mechanics. Indeed, in Conclusions, some quantum effects including the Hawking radiation are mentioned. Is there any possibility to acquire any clue to examine the quantum physics of black holes? 

4. It is understood that this article is a kind of communication. Indeed, however, if it is possible, more related astrophysical investigations including perspective and outlook for related studies to verify that the analogy argued here can be verified based on fundamental physics. 

5. It is finally recommended that the wordings and grammar of English should be checked throughout the present manuscript.

Round 2

Reviewer 2 Report

The authors have addressed all the points in the current manuscript and therefore recommend publication. 

Reviewer 3 Report

The authors' answers to the review report are appreciated very much. 
In the revised manuscript, the points suggested in the review report 
have been reconsidered. Thus, this paper can be accepted for publication 
in Universe.